# ESD Ideas: A Weak Positive Feedback Between Sea Level and the Planetary Albedo

Ben Marzeion[1]

[1]Institute of Geography and MARUM - Center for Marine Environmental Sciences, University of Bremen, Bremen, Germany

**Correspondence:** Ben Marzeion (ben.marzeion@uni-bremen.de)

**Abstract.** Since the planetary albedo of Earth above ocean is typically lower than above land surface, increasing sea-level reduces the planetary albedo. This causes a feedback that is very weak on the global scale, but significantly positive. Its amplitude can be assumed to be higher locally and to grow with the considered time scale.

On multi-centennial and longer time scales, sea-level change is projected (e.g., Levermann et al., 2013; DeConto and Pol-
lard, 2016; van Breedam et al., 2020) and reconstructed (e.g., Foster and Rohling, 2013; Lambeck et al., 2014; Dutton et al., 2015) to exceed several meters per Kelvin of global mean temperature change. In the absence of coastal protection measures, multi-meter changes of sea level would impact the land-sea area distribution of Earth notably, since a relatively large fraction of the solid Earth surface is found close to present-day sea level.

Fig. 1a shows the climatological annual mean, zonally averaged planetary albedo, obtained from the Clouds and the Earth's Radiant Energy System (CERES, Loeb et al., 2018), where every pixel has been classified as either completely above land surface, completely above ocean, or including a coastline, based on the coastline data set from Wessel and Smith (1996). In the high latitudes, the differences in albedo above land and ocean surfaces are small but show large seasonal variability due to changing cloud, ice and snow cover. But particularly in the subtropics, where cloudiness is usually low, the planetary albedo of
Earth is higher above land surfaces than above the ocean. This pattern indicates the possibility of a positive feedback between sea level and planetary albedo.

To obtain a first rough estimate of the magnitude of this feedback, we rely on projected sea-level rise, since the land sur-
face elevation is known in much higher resolution and accuracy than the coastal bathymetry. Here, we use the estimate of the
regional patterns of sea-level change from Levermann et al. (2013) for temperature increases between 0 and 5 K above pre-industrial temperatures, for a time scale of 2000 years. Note, however, that the magnitude of sea-level change depends strongly on the considered time scale (e.g., van Breedam et al., 2020), such that the estimate of the feedback strength will be strongly influenced by the choice of considered time scale.

For every $1 \times 1$ degree pixel of the planetary albedo from CERES, and for different global mean temperatures, we estimate the increase in the fraction of pixel area that is below local sea level. We base this estimate on the SRTM surface elevation data

set (Farr et al., 2007) with 3 arc-seconds resolution between 60°S and 60°N, and the ETOPO1 data set (Amante and Eakins, 2009) with 1 arc-minute resolution for the rest of the globe. The land surface is assumed to be submerged if the elevation is below the local (interpolated by nearest-neighbor) relative sea-level rise. Figure 1b shows the result of this analysis for a global mean temperature anomaly $\Delta T = 3$ K. To estimate the resulting change of the planetary albedo, we increase the ocean area of each latitudinal band by the area of submerged land, and decrease the land surface area by the same amount. We assign the climatological monthly mean, zonal mean albedo of ocean pixels to the new ocean area and assume that the submerged land had the climatological monthly mean, zonal mean albedo of land surface pixels. This corresponds to a widening of the ocean and narrowing of the land within the latitudinal band, while assuming that coastal areas shift in location, but do not change their albedo. As a result, we obtain estimates of zonal mean albedo change induced by sea-level rise for every month, and for the different temperature anomalies. From this albedo change, we calculate the associated radiative forcing change using the incoming solar radiation (also obtained from CERES).

Figure 1c shows the annual mean of this radiative forcing change as a function of latitude and $\Delta T$; Fig. 1d shows the seasonal evolution at $\Delta T = 3$ K. For the majority of latitudes, the resulting radiative forcing is negligible. There are very few latitudes for which we estimate a weak negative forcing change. Just south of the equator, in the northern subtropics, and in the northern subpolar latitudes, we obtain a positive radiative forcing change. The radiative forcing change in the Southern Hemisphere is weaker than in the Northern Hemisphere because there is less coastline that can be affected by sea-level change. There is considerable seasonal variability in the forcing change, and the overall amplitude of the signal increases with $\Delta T$, as is to be expected. For $\Delta T > 4$ K, the forcing exceeds 0.5 Wm$^{-2}$ locally.

Figure 1e shows the global mean, annual mean forcing change as a function of the radiative forcing required to achieve the considered $\Delta T$, assuming a climate sensitivity of 0.8 K/(Wm$^{-2}$). The response is almost linear, and corresponds closely to an amplification of 1 % of the radiative forcing. This implies that the feedback on the global scale and the time scale of 2000 years is very weak. However, an uncertainty estimate obtained by considering the uncertainty of projected sea-level rise from Levermann et al. (2013) indicates that it significantly positive (shading in Fig. 1e).

We have only included uncertainty of the projected sea-level rise in the uncertainty assessment. However, there is reason to assume that this is the most significant source of uncertainty: due to the zonal averaging, the uncertainty of surface elevation (Farr et al., 2007; Amante and Eakins, 2009) can be considered negligible. The repetition of the analysis with albedo data from the Earth Radiation Budget Experiment (ERBE, Barkstrom et al., 1990) yields very similar results (the amplification factor is slightly lower, but well within our uncertainty estimate).

By basing the analysis on the climatology of observed albedo, we isolated the potential impact of sea-level change, ignoring other considerably stronger albedo effects associated with climate change (e.g., Flanner et al., 2011; Zelinka et al., 2017; Wunderling et al., 2020). Since high-resolution bathymetry data are not available, a potentially negative contribution to the

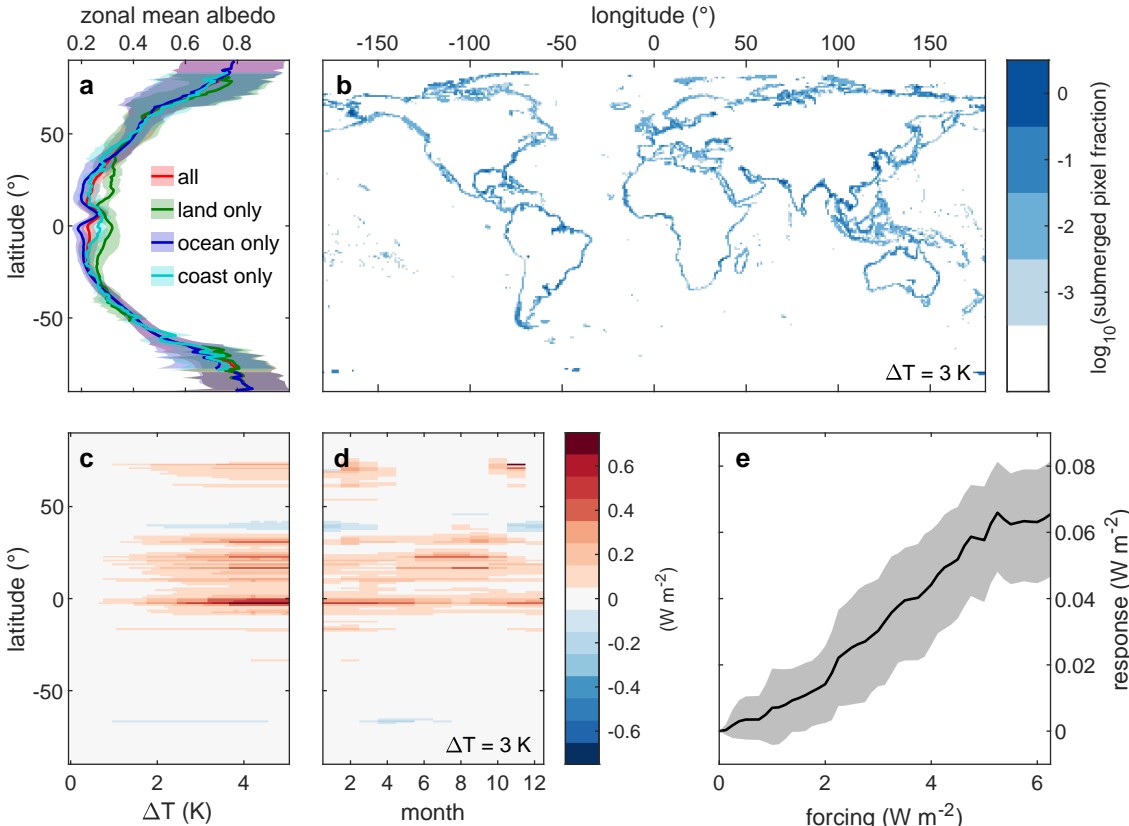

**Figure 1. a:** Annual mean, zonal mean planetary albedo as a function of latitude, for the different classes of pixels (see text). Shading indicates one standard deviation of the monthly means. **b:** Increase in the fraction of pixel area that is below local sea level at $\Delta T = 3$ K. **c:** Zonal mean, annual mean radiative forcing change resulting from sea-level induced albedo change as a function of $\Delta T$ and latitude. **d:** As c, but as a function of month and latitude, at $\Delta T = 3$ K. **e:** Global mean, annual mean response of the forcing resulting from sea-level induced albedo change as a function of the forcing required to reach the given $\Delta T$; black line indicates best estimate, gray shading indicates one standard error.

feedback by sinking relative sea level in the near field of the Antarctic and Greenland ice sheets cannot be considered here. It should also be pointed out that there is no reason to assume that the relatively constant amplification factor of 1 % will hold over all temperature anomalies. Instead, it might be speculated that albdeo changes caused by lower sea levels during large-scale glaciations could have been considerably larger, when large areas of present-day shelf seas were dry. Köhler et al. (2010) roughly estimated this effect to 0.6 W/m$^{-2}$ for the last glacial maximum. Similarly, the effect may be larger in the long-term future, due to larger changes in sea level (van Breedam et al., 2020).

Overall, our results suggest a potential role of a feedback between sea level and planetary albedo on multi-millennial time scales, significantly faster than the multi-million year time scales where the tectonic redistribution of land surface has been considered to be relevant since many years (e.g., Barron et al., 1980; Burrett, 1982).

*Data availability.* All data used in the analysis are available from the cited references.

*Competing interests.* The author declares no competing interests.

*Acknowledgements.* The author acknowledges support by the German Research Foundation, grant MA 6966/1-2. I am grateful to the two anonymous reviewers for their helpful comments.

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
