# Peer review of "ESD Ideas: A Weak Positive Feedback Between Sea Level and the Planetary Albedo"

_Earth System Dynamics, 2021_

## Author Response (AR1)

**1 Response to Reviewer 1**

I would like to thank the reviewer for their thoughtful and constructive comments.

**Major issues**:

1. **Comment:** My main concern is that for the land-surface albedo a zonally averaged value is used (lines 29-31) whereas it could be expected that what actually matters is the albedo of coastal areas. Because the albedo considerably differs between the various types of land surfaces, the difference between the zonally averaged and coastal values is not necessarily negligible. This certainly merits discussion.

   **Response:** I completely agree with the reviewer's argument that albedo differs strongly between various types of land surface, and that the difference between the zonally averaged and coastal values is not necessarily negligible. This is the reason why I chose to NOT change the albedo just of the coastal pixels. Following this idea, the albedo of former land pixels would have to be changed into that of coastal pixels, and we don't know where exactly this would happen. By using the zonally averaged values, I "widen" the ocean and "narrow" the land areas by the appropriate amount. In order to clarify, I will add the following sentence to the revised manuscript: "This corresponds to a widening of the ocean and narrowing of the land within the latitudinal band, while assuming that coastal areas shift in location, but do not change their albedo."

2. **Comment:** Another point is the lack of interpretation of the results obtained. Non-expert readers would be intrigued by the small differences between the land and ocean albedo at high latitudes (Fig. 1a), by the pixel-including-coast albedo being smaller in subtropics than in the equatorial region (Fig. 1a), and by the radiative forcing that is larger in the northern hemisphere than in the southern hemisphere (Figs. 1c and 1d). This, I guess, could be related to the influence of snow and ice cover, to the effect of clouds, and to the distribution of land and ocean between the two hemispheres. Again, a discussion would be of interest.

   **Response:** In the revised manuscript, I will add the following sentences in the appropriate sections:

   "In the high latitudes, the differences in albedo above land and ocean

surfaces are small but show large seasonal variability due to changing cloud, ice and snow cover."

"The radiative forcing change in the Southern Hemisphere is weaker than in the Northern Hemisphere because there is less coastline that can be affected by sea-level change."

Unfortunately, a more comprehensive discussion is hardly possible in the short format of an "ESD Ideas" manuscript.

**Technical corrections**:

1. **Comment:** Line 6: 'Meters' should be substituted by 'meters'
   **Response:** Will be corrected in the revised manuscript.

2. **Comment:** Line 12: 'coast a line' should presumably be substituted by 'a coastline'
   **Response:** Will be corrected in the revised manuscript.

3. **Comment:** Line 17: 'know' should be substituted by 'known'
   **Response:** Will be corrected in the revised manuscript.

**2 Response to Reviewer 2**

I would like to thank the reviewer for their comments and suggestion.

1. **Comment:** The only thing that the author could add is a brief discussion of the paper by N. Wunderling, M. Willeit, J.F. Donges, and R. Winkelmann: Global warming due to loss of large ice masses and Arctic summer sea ice, Nature Communications (2020).
   **Response:** In the revised manuscript, I will include this reference in the discussion where I point out other, likely stronger feedbacks between climate and albedo. I will also add a citation of Koehler et al. (2010, Quaternary Science Reviews 29, pp. 129-145) who estimated the albedo change caused by the sea-level drop during the LGM.